# Interferon Alpha Therapy Increases Pro-Thrombotic Biomarkers in Patients with Myeloproliferative Neoplasms

**DOI:** 10.3390/cancers12040992

**Published:** 2020-04-17

**Authors:** Dorothée Faille, Lamia Lamrani, Stéphane Loyau, Marie-Geneviève Huisse, Marie-Charlotte Bourrienne, Sawsaneh Alkhaier, Bruno Cassinat, Yacine Boulaftali, Jérôme Debus, Martine Jandrot-Perrus, Christine Chomienne, Christine Dosquet, Nadine Ajzenberg

**Affiliations:** 1INSERM UMR_S1148, Université de Paris, CEDEX 18, F-75877 Paris, France; lamia.lamrani@hotmail.fr (L.L.); stephane.loyau@inserm.fr (S.L.); marie-genevieve.huisse-ext@aphp.fr (M.-G.H.); marie-charlotte.bourrienne@inserm.fr (M.-C.B.); yacine.boulaftali@inserm.fr (Y.B.); martine.jandrot-perrus@inserm.fr (M.J.-P.); nadine.ajzenberg@aphp.fr (N.A.); 2Laboratoire d’Hématologie, AP-HP, Hôpital Bichat, CEDEX 18, F-75877 Paris, France; jerome.debus@aphp.fr; 3Service de Biologie Cellulaire, AP-HP, Hôpital Saint Louis, CEDEX 10, F-75475 Paris, France; salkhaeir@ch-valence.fr (S.A.); bruno.cassinat@aphp.fr (B.C.); christine.chomienne@aphp.fr (C.C.); christine.dosquet@aphp.fr (C.D.); 4INSERM UMR_S1131, Université de Paris, F-75010 Paris, France; 5Laboratoire d’Hématologie, AP-HP, Hôpital Louis Mourier, CEDEX, F-92701 Colombes, France

**Keywords:** myeloproliferative neoplasms, interferon, prothrombotic markers

## Abstract

Myeloproliferative neoplasms (MPN) are associated with an increased risk of arterial and venous thrombosis. Pegylated-interferon alpha (IFN) and hydroxyurea (HU) are commonly used to treat MPN, but their effect on hemostasis has not yet been studied. The aim of our study was to determine whether IFN and HU impact the biological hemostatic profile of MPN patients by studying markers of endothelial, platelet, and coagulation activation. A total of 85 patients (50 polycythemia vera and 35 essential thrombocythemia) were included: 28 treated with IFN, 35 with HU, and 22 with no cytoreductive drug (non-treated, NT). Von Willebrand factor, shear-induced platelet aggregation, factor VIII coagulant activity (FVIII:C), fibrinogen, and thrombin generation with and without exogenous thrombomodulin were significantly higher in IFN-treated patients compared to NT patients, while protein S anticoagulant activity was lower. In 10 patients in whom IFN therapy was discontinued, these hemostatic biomarkers returned to the values observed in NT patients, strongly suggesting an impact of IFN therapy on endothelial and coagulation activation. Overall, our study shows that treatment with IFN is associated with significant and reversible effects on the biological hemostatic profile of MPN patients. Whether they could be associated with an increased thrombotic risk remains to be determined in further randomized clinical studies.

## 1. Introduction

Myeloproliferative neoplasms (MPN) are clonal diseases resulting from the dysregulated production of mature myeloid cells by hematopoietic stem cells affected by a molecular defect. BCR-ABL-negative MPN include polycythemia vera (PV), essential thrombocytemia (ET), and primary myelofibrosis. They are associated with somatic mutations in 3 driver genes: JAK2, CALR, and MPL in more than 90% of patients. The most frequent one is JAK2V617F mutation [1]. Irrespective to the mutation, the hyperactivation of JAK2-STAT signaling has a central role in MPN pathophysiology [2,3]. MPN are characterized by a predisposition to thrombotic events that significantly affect patient morbidity and mortality. Thrombotic events at diagnosis and during follow up are more frequent in PV than in ET, with the prevalence of major thrombosis at diagnosis ranging from 19% to 38% and from 7% to 26%, respectively [4]. In both PV and ET, arterial thrombosis accounts for approximately two-thirds of events and consists mainly in ischemic stroke and in myocardial infarction. Deep vein thrombosis, pulmonary embolism, and splanchnic vein thrombosis are the most frequent localizations of venous events. The pathogenesis of thrombosis is assumed to be multifactorial. An increased number and abnormal functions of blood cells both contribute to the prothrombotic phenotype. Neoplastic cells produce cytokines that are responsible for the proinflammatory phenotype of vascular endothelial cells, contributing to thrombotic risk [5]. JAK2V617F expression by endothelial cells could also alter their phenotype [6]. Thus, numerous studies have provided evidence that the pathogenesis of MPN-associated thrombosis involves the activation of platelets (including the production of procoagulant microvesicles), of neutrophils (including the release of proteolytic enzymes and reactive oxygen species), of red blood cells and of endothelial cells [7].

Risk stratification in PV and ET has been designed to estimate the likelihood of thrombotic complications [8]. Predictors of arterial complications include age >60 years, leukocytosis >11 G/L, prior history of thrombosis, and cardiovascular risk factors [9]. High-risk PV and ET patients require cytoreductive therapy, while low-risk patients require daily aspirin therapy [8]. Treated PV patients with hematocrit less than 45% have a lower risk of cardiovascular death and major thrombosis than patients with higher hematocrit [10]. The first-line drug of choice for cytoreductive therapy, in both PV and ET, is hydroxyurea (HU), while the second-line drugs of choice include pegylated-interferon alpha (IFN) [8].

The IFN mechanism of action in MPN has not been completely elucidated. IFN has a direct pro-apoptotic effect on myeloid progenitors and more particularly on erythroid progenitors in humans as well as in mouse models. A direct anti-proliferative effect of IFN on stem cells has also been suggested as well as an immunomodulator effect that enhances the host immune response and restores efficient immune surveillance. However, there is no clear demonstration of such mechanisms in human MPN [11]. Interestingly, Prendergast and colleagues described that IFN treatment of mice led to a rapid stimulation of bone marrow endothelial cells in vivo [12].

Unexpectedly, HU, but not IFN, was recently reported to enhance the expression of adhesion molecules at the surface of red blood cells from PV patients, resulting in increased adhesion to laminin [13]. Thus, it is important to better characterize the effect of both drugs with respect to the activation of blood cells and coagulation.

Therefore, we evaluated biological markers of coagulation and endothelial activation in MPN patients who were treated by IFN compared to patients treated by HU and to patients with no cytoreductive drugs at the moment of inclusion.

## 2. Results

### 2.1. Patient Characteristics

Demographic, clinical, and biological characteristics of patients are depicted in Table 1. Among the 85 patients (50 PV and 35 ET), 22 were not treated with any cytoreductive drug (NT) at the inclusion, 35 were treated by HU and 28 by IFN. HU-treated patients were older than NT and IFN-treated patients (*p* = 0.001 and 0.0002, respectively). PV was more frequent in HU-and IFN-treated patients than in the NT group (*p* = 0.002). Median time from diagnosis to inclusion was 7 years in NT group, 3 years in HU-treated group, and 10.5 years in IFN-treated group. Cytoreductive therapy had been initiated at diagnosis in 23/35 (65.7%) and in 8/28 (28.6%) patients from HU- and IFN-treated groups, respectively. Median time from initiation of treatment to inclusion in the study was not statistically different between patients treated with HU or IFN. All these patients were treated for at least 4 months and were receiving aspirin when included. The median cumulative dose of IFN in IFN-treated patients was 10,800 µg (interquartile range, IQR: 6480–16,470). Four patients with PV were treated by phlebotomy. Among them, one was also treated by HU. The occurrence of previous thrombotic events (arterial and venous) was similar between the 3 groups of patients. Bleeding episodes such as epistaxis or bruises were not statistically different according to treatment. Frequency of the JAK2V617F mutation was in agreement with previous published data (51% in ET and 98% in PV patients) (Appendix A) [1,14] and did not differ among the NT, HU-treated, and IFN-treated patients. JAK2V617F burden was not statistically different among the 3 groups of patients according to treatment (mean ± SD: 19 ± 20%, 34 ± 22%, and 29 ± 25% in NT, HU-treated, and IFN-treated patients, respectively). As expected, platelet count was higher in ET compared to PV patients (*p* < 0.0001), and hematocrit and hemoglobin were higher in PV compared to ET patients (*p* = 0.001 and *p* = 0.01, respectively) (Appendix A). Blood cell counts were higher in NT patients compared to treated patients. Platelet count was higher in NT patients compared to HU- and IFN-treated patients (*p* = 0.007 and *p* = 0.0003, respectively). HU-treated patients had also a significant higher platelet count than IFN-treated patients (*p* = 0.005). Hemoglobin was higher in NT and HU-treated patients compared to IFN-treated patients (*p* = 0.01 and *p* = 0.03, respectively) and leukocyte count was significantly higher in NT and HU-treated patients compared to IFN-treated patients (*p* = 0.0001 and *p* = 0.0003, respectively).

### 2.2. Markers of Endothelial Activation Are Increased in IFN-Treated Patients

To analyze the effect of IFN treatment on the activation of the vascular endothelium, we measured the levels of von Willebrand factor (vWF) that is released from endothelial cells upon activation. vWF antigen was higher in IFN-treated patients compared to NT and HU-treated patients (*p* < 0.0001 and *p* = 0.004, respectively) (Table 2 and Figure 1A). The same results were observed for and vWF activity (Table 2 and Figure 1B). In a logistic regression analysis, these results remained significant after adjustment for age, sex, and disease type (ET or PV). In HU-treated patients, vWF antigen and activity were also significantly higher compared to NT patients (*p* = 0.002 and *p* = 0.03, respectively). No correlation could be observed between IFN dose and vWF antigen or activity. As platelet aggregation under high shear conditions is dependent of vWF, we evaluated shear-induced platelet aggregation (SIPA) at 4000 s^−1^ in all patients. SIPA was higher in IFN-treated patients compared to NT and HU-treated patients (*p* = 0.0005 and *p* = 0.002, respectively), independently of age, sex, and disease type. No difference was observed between HU-treated and NT patients (Table 2 and Figure 1C).

### 2.3. IFN Treatment Does Not Activate Platelets In Vivo

To analyze the effect of IFN treatment on platelet activation in vivo, the exposure of P-selectin, a granule secretion marker, and activated GPIIbIIIa at the platelet surface was measured. No difference was observed between the 3 groups of patients, and levels remained in the normal ranges (Table 3). To further examine platelet activation status, we determined the level of circulating platelet–monocyte aggregates (PMA) and platelet–neutrophil aggregates (PNA) in the 3 groups of patients according to treatment. Percentages of aggregates remained in the normal ranges and did not differ among the 3 groups.

### 2.4. IFN Treatment Does Not Modify Biological Response to Aspirin Treatment

Since aspirin could influence the profile of platelet activation markers, we verified the compliance of patients and tested the effectiveness of aspirin in inhibiting arachidonic acid-induced platelet aggregation. A total of 12 patients were not evaluable, because of too recent aspirin intake. Biological efficacy of aspirin was not different between the 3 groups of patients according to treatment (Appendix A).

### 2.5. IFN-Treated Patients Display A Hypercoagulable Profile

To investigate the hemostatic balance between anti- and pro-coagulant parameters, we first measured pro-coagulant factors such as factor VIII coagulant activity (FVIII:C) and fibrinogen, and anticoagulant proteins such as protein S, protein C, and antithrombin. FVIII:C was significantly higher in IFN-treated patients compared to NT and HU-treated patients (*p* = 0.003 and *p* < 0.0001, respectively) (Table 4 and Figure 2A). Furthermore, fibrinogen was slightly but significantly higher in IFN-treated patients compared to NT patients (*p* = 0.004) (Table 4 and Figure 2B). Conversely, protein S was significantly lower in IFN-treated patients compared to NT patients (*p* < 0.0001) and to HU-treated patients (*p* < 0.0001) (Table 4 and Figure 2C). In a logistic regression analysis, these results remained significant after adjustment for age, sex, and disease type. Levels of protein C and antithrombin were in the normal range without difference among the 3 groups of patients (Table 4). 

Next, we used a global thrombin generation assay with or without the addition of recombinant thrombomodulin in plasma. In the absence of thrombomodulin, the thrombin peak (Table 5 and Figure 3A) and velocity index (Table 5 and Figure 3C) were significantly higher in HU- (*p* = 0.02) and in IFN-treated patients (*p* = 0.03 and *p* = 0.04, respectively) compared to NT patients. In a logistic regression analysis, these results were not significantly different anymore, suggesting that thrombin generation results were dependent of age, sex, and/or disease type. Neither thrombin peak nor velocity differed between HU- and IFN-treated patients. No difference was evidenced for the endogenous thrombin potential (ETP) among the 3 groups (Table 5 and Figure 3B). As expected, adding thrombomodulin to the plasma markedly inhibited thrombin generation as indicated by the lower peak and ETP values (Table 5 and Figure 3D–E) compared to the results obtained in the absence of thrombomodulin. In the presence of thrombomodulin, thrombin peak (Table 5 and Figure 3D), ETP (Table 5 and Figure 3E), and velocity index (Table 5 and Figure 3F) were significantly higher in HU- (*p* = 0.02, *p* = 0.04 and *p* = 0.02, respectively) and in IFN-treated patients (*p* = 0.01, *p* = 0.02 and *p* = 0.01, respectively) compared to NT patients. No difference was evidenced between HU- and IFN-treated patients (Table 5 and Figure 3D–F).

### 2.6. Discontinuation of IFN Treatment Restores Hemostatic Parameters to Normal Levels

To further validate our hypothesis that the elevation of plasma pro-coagulant biomarkers in IFN-treated patients was directly related to IFN therapy, we took advantage of the discontinuation of IFN treatment in some patients. A total of 10/28 IFN-treated patients (5 ET and 5 PV) discontinued IFN treatment because of side effects (*n* = 4), hematological or molecular response (*n* = 5), or personal preference (*n* = 1).

Biological parameters were tested at least 6 months after IFN discontinuation. Interestingly, vWF antigen and activity, FVIII:C, fibrinogen, and protein S activity returned to levels similar to those observed in NT patients and were significantly different from levels observed during IFN treatment. vWF antigen decreased significantly from 182% (153–273%) to 92% (78–104%) (*p* = 0.002), vWF activity from 172% (135–228%) to 92% (82–128%) (*p* = 0.002), FVIII:C from 158% (128–217%) to 112% (74–141%) (*p* = 0.004), fibrinogen from 3.7 (3.4–4.6) to 3.1 (2.8–3.4) g/L (*p* = 0.004), and protein S activity increased from 62% (48–75%) to 87% (77–111%) (*p* = 0.002) (Figure 4A–F).

## 3. Discussion

This observational single-center cohort study aimed to evaluate the impact of IFN and HU on the biological hemostatic profile of MPN patients. Here, we report that IFN therapy, and to a lesser extent HU treatment, are associated with significantly higher levels of pro-thrombotic markers compared to NT patients, independently of age, sex, and disease type. Most clinical studies have reported a complete and sustained molecular response in 35% of MPN patients after IFN administration [11,15], but there is currently no clinical data on the potential side effects of IFN on hemostasis.

IFN alpha is able to activate the MAP kinase pathway responsible for inflammation [16]. It has been demonstrated that there is a synergistic effect of IFN alpha and Toll-like receptor 4 ligands in the development of atherosclerotic plaque [17]. In mouse models, IFN alpha treatment led to a rapid stimulation of bone marrow endothelial cells in vivo [12], promoting inflammation. In the present study, we show that vWF antigen and activity were significantly higher in IFN-treated patients than in NT patients. These results are consistent with those reported in patients treated by IFN for hepatitis, who presented similarly higher vWF levels, suggesting an activation of endothelial cells in vivo [18].

SIPA in high shear rate conditions is able to reproduce stenotic arteries and is usually correlated to vWF level [19]. Consistent with the high levels of vWF found in our study, SIPA was also higher in IFN-treated patients. We also evaluated platelet activation markers. The exposure of P-selectin and activated GPIIbIIIa as well as platelet–leucocyte aggregate levels remained low and were similar among groups. Since the frequency of biological resistance to aspirin was comparable among the 3 treatment groups, we can assume that these results were not influenced by aspirin. Overall, these data are not in favor of in vivo platelet activation in our cohort and are discrepant with data from other studies reporting elevated markers of platelet activation in MPN patients [20]. This could be explained by two hypotheses: (i) NT patients had a low level of disease progression and (ii) cytoreductive therapy was efficient in the treated groups.

Aside from vWF, FVIII is the only coagulation factor upstream of thrombin generation that was higher in IFN-treated patients and to a lesser extent in HU-treated patients compared to NT patients, probably because FVIII circulates associated to vWF. Interestingly, fibrinogen level was also significantly higher in these patients, suggesting an inflammatory state. However, C-reactive protein was not statistically different among groups, which was probably because of its shorter half-life as compared to fibrinogen. We reported an elevation of thrombin generation in HU- and IFN-treated patients compared to NT patients. Similar results were observed in a previous study comparing PV, ET, and controls [21]. In two other studies, a decrease in thrombin generation in MPN patients compared to controls was observed but with much higher tissue factor concentrations (up to 6.8 pM) [22,23]. In order to assess the effectiveness of the protein C/protein S anticoagulant pathway, thrombomodulin was added to the thrombin generation assay [24]. The elevation of thrombin generation in HU- and IFN-treated patients was even more significant in the presence of thrombomodulin. This could be explained by the reduced levels of free protein S as previously reported in a global coagulation study in MPN patients compared to controls [25]. These data suggest a pro-coagulant imbalance in IFN-treated patients compared to HU-treated and NT patients, even though we could not find any correlation between protein S levels and thrombin generation parameters in the presence of thrombomodulin, which was possibly due to the limited number of patients studied.

Importantly, we found that vWF antigen and activity, protein S activity, FVIII:C, and fibrinogen returned to levels observed in NT patients, when patients interrupted IFN treatment for at least 6 months. These data support the causal effect of IFN treatment in the modification of these biological parameters and thus in the generation of a potential pro-thrombotic tendency.

This study has certain limitations in addition to the limited number of patients included. First, the patients of our cohort were in a steady state explaining why some markers such as platelet activation markers were not different between groups, in contrast to previous studies [20,21,22,23,26]. Moreover, treatment by aspirin in all patients could have minimized their platelet activation profile. Second, as only two patients presented thrombotic events after treatment initiation, the association between thrombosis and treatment could not be established. Finally, the low number of IFN-treated patients with calreticulin (CALR) mutation did not allow for subgroup analysis of pro-thrombotic biomarkers according to the mutational status in IFN-treated patients.

## 4. Materials and Methods

### 4.1. Patients

Between January 2012 and February 2014, 85 patients referred to the Unit of Cellular Biology at Saint-Louis hospital in Paris were enrolled in this study. PV and ET patients were diagnosed according to WHO criteria. Blood samples were collected in the Department of Biological Hematology in Bichat hospital. The INSERM Ethics Evaluation Committee (CEEI/IRB) approved the protocol (IRB00006477, opinion number 11–106). All patients provided a written informed consent. At the moment of inclusion, patients were receiving treatment with HU or IFN for at least 4 months or were not treated with any cytoreductive drug. None of the patients was taking anticoagulant or antiplatelet therapy other than aspirin.

### 4.2. Genotyping MPN Mutations

JAK2V617F mutation was detected using the Ipsogen JAK2 MutaQuant kit (Qiagen, Courtaboeuf, France) according to manufacturer’s instructions. A high resolution melting (HRM) method was used to detect *JAK2* exon 12 mutations as previously described [27]. Screening for the presence of CALR mutations was performed as previously described [28]. Finally, p.W515L and p.W515K MPL mutations were detected using Ipsogen MPL W515L/K MutaScreen kit (Qiagen, Courtaboeuf, France) according to the manufacturer’s instructions.

### 4.3. Platelet Rich and Poor Plasma Preparation

Blood samples were collected in vacuum tubes (Greiner-Bio One, Les Ulis, France) containing citrate (0.109 M) for platelet and coagulation studies or EDTA for blood cell count. Platelet-rich plasma (PRP) was obtained by centrifugation of whole blood at 120 g for 15 min at room temperature. Platelet-poor plasma (PPP) was isolated by the double centrifugation of blood samples at 2500 g for 15 min at 18 °C and stored at 80 °C until further analysis.

### 4.4. Von Willebrand Factor (vWF) Measurements

vWF activity was determined by vWF Ristocetin Cofactor (vWF:RCo) assay using light transmission aggregometry (ChronoLog Havertown, PA, USA). PPP was incubated with lyophilized platelets (Hyphen BioMed, Neuville sur Oise, France) and ristocetin (1.2 mg/mL) (Diagnostica Stago, Asnières, France). Results were expressed as percentage of vWF activity compared to reference plasma. vWF antigen was measured by ELISA (Asserachrom vWF antigen, Diagnostica Stago) following manufacturer’s instructions.

### 4.5. Shear-Induced Platelet Aggregation (SIPA)

Shear experiments were performed on PRP by means of a coaxial cylinder-shearing device at 4000 s^−1^ as previously described [29]. SIPA was measured on an EPICS XL flow cytometer (Beckman Coulter, Villepinte, France). The results of platelet aggregation were expressed as percentage of disappearance of single platelets (DSP): DSP = [(*n*_0_-*n*)/*n*_0_] × 100, when *n*_0_ represents the single platelet population of non-sheared control sample and *n* represents that of the sheared sample.

### 4.6. Coagulation Parameter Measurements

Fibrinogen (Dade Thrombin reagent, Siemens, Marburg, Germany), factor VIII coagulant (FVIII:C) (Factor VIII deficient plasma, Siemens), protein S activity (Staclot Protein S, Diagnostica Stago), protein C anticolagulant activity (Cryocheck Clot C, Cryopep, Montpellier, France), and antithrombin (Stachrom AT III, Diagnostica Stago) were measured on a STA-R Analyser (Diagnostica Stago). 

### 4.7. Platelet Activation Markers

In vivo platelet activation was evaluated by measuring the surface expression of P-selectin (CD62P) and activated GPIIb/GPIIIa (PAC-1) by flow cytometry. Briefly, diluted whole blood was incubated with fluorescein isothiocyanate (FITC)-conjugated anti-CD62P or PAC-1 (Beckman Coulter, Villepinte, France) for 30 min in the dark. Samples were analyzed with an EPICS XL flow cytometer. Platelets were gated on forward and side scatter and the percentage of P-selectin and PAC-1-positive platelets were obtained after subtracting the percentage of positive platelets obtained with the FITC-conjugated isotype control for CD62P (IgG1) and PAC-1 (IgM), respectively.

### 4.8. Platelet–Leukocyte Complexes

Whole-blood platelet–leukocyte complexes were counted by flow cytometry. Briefly, PMA were defined as the proportion of monocytes labeled with anti-CD14-FITC (Beckman Coulter) positive for phycoerythrin (PE)-conjugated anti-CD41 (Beckman Coulter) and PNA were defined as the proportion of neutrophils labeled with phycoerythrin-cyanine 5(PC5)-conjugated anti-CD45 (Beckman Coulter) positive for anti-CD41-PE. Samples were analyzed with an EPICS XL flow cytometer.

### 4.9. Thrombin Generation Assay

Thrombin generation was measured on PPP using calibrated automated thrombogram (CAT, Diagnostica Stago) as previously described [30]. In PPP, thrombin generation was triggered by 1 pM tissue factor and 4 mM phospholipids (PPP-Reagent LOW, Diagnostica Stago) in the absence or in the presence of rabbit thrombomodulin (American Diagnostica, 4 nM). The following parameters were quantified: peak (maximal thrombin concentration, in nM of thrombin), velocity (slope of thrombin generated in nM of thrombin/min), and ETP (total amount of thrombin generated, in nM of thrombin/min).

### 4.10. Evaluation of Biological Efficacy of Aspirin

Arachidonic acid-induced platelet aggregation was studied in PRP by light transmittance aggregometry after the addition of 1.25 μM arachidonic acid (Chrono-Par, Chrono-Log corp. Havertown, PA, USA). Resistance to aspirin was defined by maximal platelet aggregation >20% tested at trough level just before the daily intake of aspirin. Platelet count in PRP was adjusted to 600 G/L with PPP if the platelet count in native PRP was above.

### 4.11. Statistical Analysis

Categorical variables are presented as number with percentage. Continuous variables were tested for normality distribution with the Kolmogorov–Smirnov test and are expressed median with IQR (25th–75th percentiles). Comparisons among the 3 groups were performed using Kruskal–Wallis test for continuous variables and Chi-square test for categorical variables. Post-hoc pairwise comparison of parameters was performed using the Mann–Whitney test and logistic regression analysis was used to adjust for age, sex, and disease type (PV or ET). A comparison of biological parameters during IFN treatment and after IFN discontinuation was performed with the Wilcoxon matched pairs test. Analysis of correlation was performed using Spearman’s rank correlation coefficient. For all of the statistical analyses, a value of *p* < 0.05 was considered statistically significant. GraphPad Prism 5 (GraphPad Software) was used for all analyses.

## 5. Conclusions

Overall, our study shows that the treatment with IFN is associated with significant and reversible effects on the biological hemostatic profile of MPN patients. The elevation of pro-thrombotic biomarkers in MPN patients treated with IFN should be considered at least a biological side effect of this treatment. Thus, our study points out that caution should be taken and further preventive antithrombotic therapy other than aspirin alone could be required in IFN-treated patients with additional common thrombotic risk factors such as FV Leiden or FII G20210A mutation and/or with cardiovascular risk factors. Indeed, previous data from an open-label trial showed an elevated number of thrombotic events in IFN-treated patients [31]. Randomized studies are required to determine the incidence of thrombosis in PV and ET patients treated with IFN compared to HU.

## Figures and Tables

**Figure 1 cancers-12-00992-f001:**
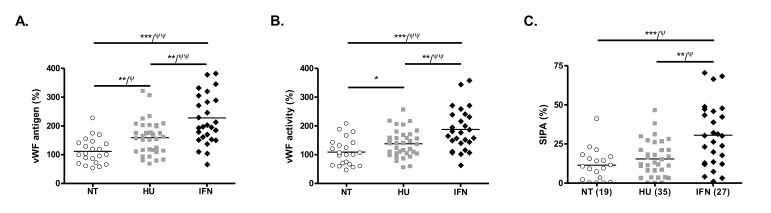
Von Willebrand factor antigen (**A**), von Willebrand factor activity (**B**), and platelet aggregation induced by high shear at 4000 s^−1^ (SIPA, **C**) in NT, HU-, and IFN-treated patients. Post-hoc pairwise comparisons of parameters were performed using Mann–Whitney test (* *p* < 0.05, ** *p* < 0.01, *** *p* < 0.001) and logistic regression analysis adjusted on age, sex, and disease type (ET or PV) (^Ψ^
*p* < 0.05, ^ΨΨ^
*p* < 0.01).

**Figure 2 cancers-12-00992-f002:**
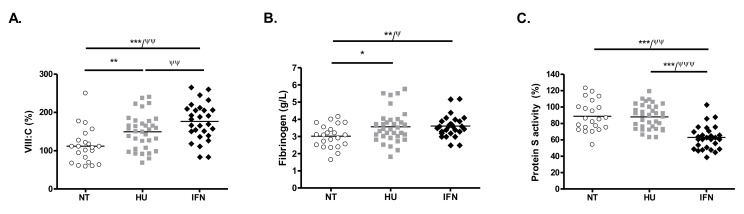
Levels of VIII:C (**A**) fibrinogen (**B**) and protein S activity (**C**) in NT, HU- and IFN-treated patients. Post-hoc pairwise comparisons of parameters were performed using the Mann–Whitney test (* *p* < 0.05, ** *p* < 0.01, *** *p* < 0.001) and logistic regression analysis adjusted for age, sex, and disease type (ET or PV) (^Ψ^
*p* < 0.05, ^ΨΨ^
*p* < 0.01, ^ΨΨΨ^
*p* < 0.001).

**Figure 3 cancers-12-00992-f003:**
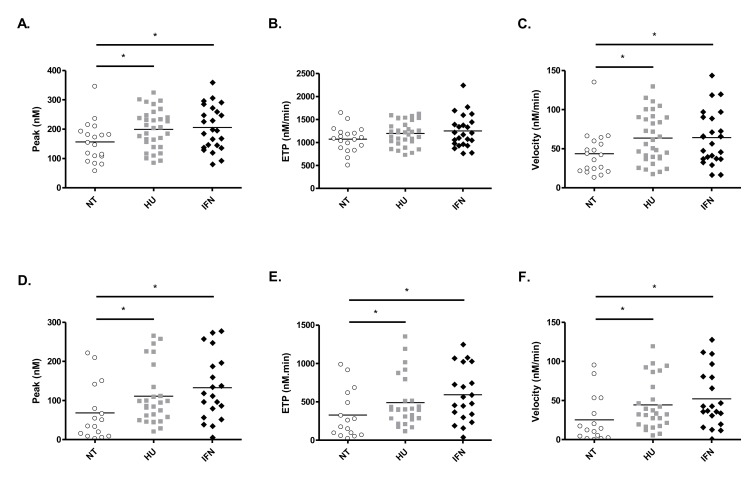
Thrombin generation in the plasma of NT, HU- and IFN-treated patients in the absence (**A**–**C**) and in the presence (**D**–**F**) of thrombomodulin. Peak of thrombin (**A**,**D**), ETP (**B**,**E**), and velocity (**C**,**F**). Post-hoc pairwise comparisons of parameters were performed using the Mann–Whitney test. * *p* < 0.05.

**Figure 4 cancers-12-00992-f004:**
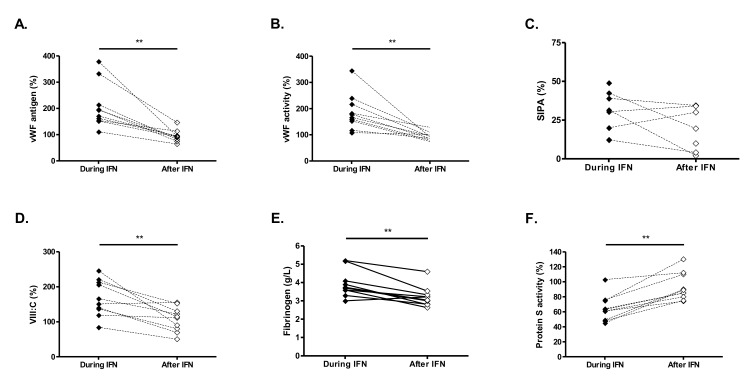
von Willebrand factor antigen (**A**), von Willebrand factor activity (**B**), platelet aggregation induced by high shear at 4000s^−1^ (SIPA, **C**), VIII:C (**D**) fibrinogen (**E**) and protein S activity (**F**) during IFN treatment and after IFN discontinuation in 10 patients. Comparison between the 2 groups was performed using Wilcoxon matched pairs test. ** *p* < 0.01.

**Table 1 cancers-12-00992-t001:** Demographic, clinical, and biological characteristics of the study population.

Characteristics	NT (*n* = 22)	HU (*n* = 35)	IFN (*n* = 28)	*p*-Value
Age, years	55.5 (50–68.5)	72 (62–78)	58.5 (50–66.5)	0.0001
Gender, male	14 (64)	16 (46)	9 (32)	0.08
Disease type				
PV	6 (27)	24 (69)	20 (71)	0.002
ET	16 (73)	11 (31)	8 (29)	
Time, years:				
from diagnosis to inclusion	7 (1–12)	3 (1–11)	10.5 (5–21.5)	0.007
from diagnosis to treatment	NA	0 (0–1)	7.5 (1–17.5)	<0.0001
from treatment to inclusion	NA	3 (0–6)	2 (1.5–3)	0.5
Aspirin dose				
75 mg	10 (45.5)	24 (68.5)	22 (79)	0.1
100 mg	11 (50)	9 (26)	5 (18)	
160 mg	1 (4.5)	2 (5.5)	1 (3)	
History of thrombosis				
Arterial	6 (27)	8 (23)	4 (14)	0.5
Venous	2 (9)	4 (11)	1 (3)	0.5
History of bleeding	0 (0)	5 (14)	7 (25)	0.04
Mutations				
JAK2V617F	15 (68)	29 (83)	23 (82)	0.4
CALR	3 (14)	5 (14)	3 (11)	
JAK2 exon 12	1 (4)	0 (0)	0 (0)	
MPL	0 (0)	0 (0)	1 (3.5)	
Triple negative	3 (14)	1 (3)	1 (3.5)	
Blood cell counts				
Platelets, 10^9^/L	552 (265–732)	324 (242–437)	194 (151–323)	0.0001
Hemoglobin, g/dL	14.7 (13.4–15.8)	14.5 (13.7–15.7)	13.7 (13–14.5)	0.03
Hematocrit, %	44.1 (39.3–46.1)	44.1 (40.9–46.2)	41.0 (38.1–43.0)	0.048
Leukocytes, 10^9^/L	6.8 (5.9–8.6)	6.1 (5.1–7.5)	4.5 (3.3–5.9)	<0.0001

ET: essential thrombocythemia, HU: hydroxyurea, IFN: interferon-treated patients, NA: not applicable, NT: not treated with a cytoreductive drug at the inclusion, PV: polycythemia vera. Results are presented as number (percentage) for categorical variables or as median (IQR) for continuous variables; *p*-value for Chi-square, Mann–Whitney or Kruskal–Wallis test.

**Table 2 cancers-12-00992-t002:** Biomarkers of endothelial activation according to cytoreductive treatment group.

Biomarkers	NT (*n* = 22) *	HU (*n* = 35)	IFN (*n* = 28) *	*p*-Value
vWF antigen, %	101 (70–143)	161 (113–202)	201 (158–298)	< 0.0001
vWF activity, %	103 (64–141)	139 (104–168)	178 (141–240)	0.0001
SIPA, % *	10.6 (2.0–16.7)	13.0 (5.8–23.5)	30.4 (13.0–46.1)	0.0004

ET: essential thrombocythemia, HU: hydroxyurea, IFN: interferon-treated patients, NT: not treated with a cytoreductive drug at the inclusion, PV: polycythemia vera, SIPA: shear-induced platelet aggregation, vWF: von Willebrand factor. Results are presented as median (IQR); *p*-value for Kruskal–Wallis test. * Due to missing points, SIPA results from 19 NT and 27 IFN-treated patients are presented.

**Table 3 cancers-12-00992-t003:** Markers of in vivo platelet activation according to cytoreductive treatment group.

Markers	NT (*n* = 22)	HU (*n* = 35)	IFN (*n* = 28)	*p*-Value
P-selectin, %	1.2 (0.4–1.6)	1.2 (0.4–2.0)	0.7 (0.3–1.6)	0.6
Activated GpIIbIIIa, %	0.4 (0.1–2.8)	1.8 (0.6–5.6)	1.2 (0.4–3.4)	0.055
PMA, %	6.8 (5.2–10.2)	9.0 (6.4–15.0)	8.0 (6.4–9.4)	0.2
PNA, %	7.8 (6.3–10.0)	9.4 (7.6–11.7)	7.5 (6.4–9.2)	0.08

ET: essential thrombocythemia, HU: hydroxyurea, IFN: interferon-treated patients, NT: not treated with a cytoreductive drug at the inclusion, PMA: platelet–monocyte aggregates, PNA: platelet–neutrophil aggregates, PV: polycythemia vera. Results are presented as median (IQR); *p*-value for Kruskal–Wallis test.

**Table 4 cancers-12-00992-t004:** Pro- and anti-coagulant factors according to cytoreductive treatment group.

Factors	NT (*n* = 22)	HU (*n* = 35)	IFN (*n* = 28)	*p*-Value
VIII:C, %	109 (68–136)	155 (107–178)	176 (145–208)	<0.0001
Fibrinogen, g/dL	3.1 (2.4–3.6)	3.4 (3.0–3.9)	3.6 (3.2–3.9)	0.02
Protein S, %	85 (73–104)	90 (73–100)	62 (50–70)	<0.0001
Protein C, %	103 (96–123)	124 (101–130)	117 (93–128)	0.4
Antithrombin, %	108 (101–111)	106 (101–118)	111 (103–119)	0.3

ET: essential thrombocythemia, HU: hydroxyurea, IFN: interferon-treated patients, NT: not treated with a cytoreductive drug at the inclusion, PV: polycythemia vera. Results are presented as median (IQR); *p*-value for Kruskal–Wallis test.

**Table 5 cancers-12-00992-t005:** Parameters of thrombin generation according to cytoreductive treatment group.

Parameters	NT (*n* = 22)	HU (*n* = 35)	IFN (*n* = 28)	*p*-Value
Without thrombomodulin				
Peak, nM	154 (107–192)	205 (147–243)	197 (142–266)	0.04
ETP, nM.min	1080 (879–1218)	1215 (1021–1428)	1194 (975–1422)	0.2
Velocity, nM/min	42 (21–60)	58 (38–90)	61 (37–90)	0.049
With thrombomodulin				
Peak, nM	42 (11–110)	89 (53–162)	115 (68–192)	0.02
ETP, nM.min	215 (78–552)	397 (274–653)	525 (321–887)	0.03
Velocity, nM/min	13.0 (3.0–43.0)	32 (18–76)	40 (25–80)	0.02

ET: essential thrombocythemia, ETP: endogenous thrombin potential, HU: hydroxyurea, IFN: interferon-treated patients, NT: not treated with a cytoreductive drug at the inclusion, PV: polycythemia vera. Results are presented as median (IQR); *p*-value for Kruskal–Wallis test.

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
