# Peer review of "Interferon Alpha Therapy Increases Pro-Thrombotic Biomarkers in Patients with Myeloproliferative Neoplasms"

_cancers, 2020, doi:10.3390/cancers12040992_

Round 1

Reviewer 1 Report

The study describes single-center cohort evaluation of impact of IFN and HU on the biological hemostatic profile of MPN patients. The study is well designed, the evaluated parameters are thoughtfully chosen and analyzed, and results are clear and well discussed. The authors pointed out, that the major limitation of the study is the number of the included patients, nevertheless the observed association of IFN treatment with elevated pro-thrombotic biomarkers is strong enough to reach statistical significance.

The paper is extremely well written with any mistakes, results are sound and nicely described. There is not much which could be done differently (taken into the account the scope of the study and its clinical background). I also believe that the evaluation of pro-thrombotic biomarkers in MPN patients should be considered in the clinics therefore it is important to publish such preliminary data and support required upcoming randomized studies to determine the incidence of thrombosis in MPN after IFN treatment.

Just less than minor points.

The only mistake I have found is missing “space” in Table 1, Aspirin dose, HU treatment 24 “space” (68.5).

Figure 1 C and Table 2 do not agree on patients included in the SIPA evaluation. Table 2: 22, 35, 28, Figure 1: 19, 35, 27.  

Author Response

Reviewer #1

The study describes single-center cohort evaluation of impact of IFN and HU on the biological hemostatic profile of MPN patients. The study is well designed, the evaluated parameters are thoughtfully chosen and analyzed, and results are clear and well discussed. The authors pointed out, that the major limitation of the study is the number of the included patients, nevertheless the observed association of IFN treatment with elevated pro-thrombotic biomarkers is strong enough to reach statistical significance.

The paper is extremely well written with any mistakes, results are sound and nicely described. There is not much which could be done differently (taken into the account the scope of the study and its clinical background). I also believe that the evaluation of pro-thrombotic biomarkers in MPN patients should be considered in the clinics therefore it is important to publish such preliminary data and support required upcoming randomized studies to determine the incidence of thrombosis in MPN after IFN treatment.

Just less than minor points.

The only mistake I have found is missing “space” in Table 1, Aspirin dose, HU treatment 24 “space” (68.5).

Response: as pointed out by the reviewer, the missing space was added in Table 1, page 3, as follows: “24 (68.5)”

Figure 1 C and Table 2 do not agree on patients included in the SIPA evaluation. Table 2: 22, 35, 28, Figure 1: 19, 35, 27.  

Response: we thank the reviewer to have highlighted this relevant discrepancy. Due to missing points, SIPA results from 19 NT-and 27 IFN-treated patients are presented. This has been specified in the legend of Table 2, page 4, line 125.

Reviewer 2 Report

Faille et al. present hemostatic profile changes after hydroxyurea (HU)or interferon alpha (IFN) treatment for Myeloproliferative neoplasms patients.  The result is potential interesting.  The pairwise comparison of during and after IFN treatment (Figure 4) strongly supports the general conclusion.

However, the general statistical analysis is problematic. The non-treated (NT), HU, and IFN groups consist both PV and ET patients that are different in hemostatic profiles. Univariate analysis is not appropriate. Patient type (PV/ET) should be considered in all analyses except pairwise comparison.

Author Response

Reviewer #2

Faille et al. present hemostatic profile changes after hydroxyurea (HU) or interferon alpha (IFN) treatment for Myeloproliferative neoplasms patients.  The result is potential interesting.  The pairwise comparison of during and after IFN treatment (Figure 4) strongly supports the general conclusion.

However, the general statistical analysis is problematic. The non-treated (NT), HU, and IFN groups consist both PV and ET patients that are different in hemostatic profiles. Univariate analysis is not appropriate. Patient type (PV/ET) should be considered in all analyses except pairwise comparison.

Response: we thank the reviewer for this relevant remark.

The comparison of hemostatic profiles between PV and ET patients has been completed in Table S1 (Supplementary materials).

Although most of the parameters that are presented in Figures 1, 2 and 3 did not differ between PV and ET, we performed a multivariate logistic regression analysis to adjust the results for age, sex and disease type (PV or ET).

This has been specified in the Statistical Analysis paragraph of the Materials and Methods section (page 11, line 344).

Figures 1 and 2 have been modified accordingly (p-values from the logistic regression analysis have been added) and Figure legends have been completed as follows: “Post-hoc pairwise comparisons of parameters were performed using Mann-Whitney test (*p< 0.05 **p<0.01 ***p<0.001) and logistic regression analysis adjusted on age, sex and disease type (ET or PV) (Yp< 0.05 YYp<0.01 YYYp<0.001)”, page 4, line 132 and page 6, line 171.

A corresponding description of the adjusted results has been added page 3, line 113 and page 5, line 160: “In a logistic regression analysis, these results remained significant after adjustment for age, sex and disease type (ET or PV)”. Page 4, line 120 and page 8, line 217 was added the following specification: “independently of age, sex and disease type”.

Only the results of thrombin generation were not significant anymore after adjustment. This has been added in the text page 6, line 176, as follows: “In a logistic regression analysis, these results were not significantly different anymore, suggesting that thrombin generation results were dependent of age, sex and/or disease type”

Round 2

Reviewer 2 Report

The authors have addressed my concerns. I have no further concern.